# Effect of Influenza Vaccination on the Reduction of the Incidence of Chronic Kidney Disease and Dialysis in Patients with Type 2 Diabetes Mellitus

**DOI:** 10.3390/jcm11154520

**Published:** 2022-08-03

**Authors:** Li-Chin Sung, Chun-Chao Chen, Shih-Hao Liu, Chun-Chih Chiu, Tsung-Yeh Yang, Cheng-Hsin Lin, Yu-Ann Fan, William Jian, Meng-Huan Lei, Hsien-Tang Yeh, Min-Huei Hsu, Wen-Rui Hao, Ju-Chi Liu

**Affiliations:** 1Division of Cardiology, Department of Internal Medicine, Shuang Ho Hospital, Taipei Medical University, New Taipei City 235, Taiwan; 10204@s.tmu.edu.tw (L.-C.S.); b101092035@tmu.edu.tw (C.-C.C.); 17257@s.tmu.edu.tw (C.-C.C.); 15535@s.tmu.edu.tw (T.-Y.Y.); runawayyu@hotmail.com (Y.-A.F.); 2Taipei Heart Institute, Taipei Medical University, Taipei 110, Taiwan; chlin@s.tmu.edu.tw; 3Division of Cardiology, Department of Internal Medicine, School of Medicine, College of Medicine, Taipei Medical University, Taipei 110, Taiwan; 4Department of Primary Care Medicine, Shuang Ho Hospital, Taipei Medical University, New Taipei City 235, Taiwan; b101103004@tmu.edu.tw; 5Graduate Institute of Medical Sciences, College of Medicine, Taipei Medical University, Taipei 110, Taiwan; 6Division of Cardiovascular Surgery, Department of Surgery, Shuang Ho Hospital, Taipei Medical University, New Taipei City 235, Taiwan; 7Division of Cardiovascular Surgery, Department of Surgery, School of Medicine, College of Medicine, Taipei Medical University, Taipei 110, Taiwan; 8Department of Emergency, University Hospitals Cleveland Medical Center, Cleveland, OH 44106, USA; william.jian@gmail.com; 9Cardiovascular Center, Lo-Hsu Medical Foundation Luodong Poh-Ai Hospital, Yilan 265, Taiwan; mhlei6401@yahoo.com.tw; 10Department of Surgery, Lotung Poh-Ai Hospital, Luodong 265, Taiwan; 836023@mail.pohai.org.tw; 11Graduate Institute of Data Science, College of Management, Taipei Medical University, Taipei 110, Taiwan; 701056@tmu.edu.tw; 12Department of Neurosurgery, Wan-Fang Hospital, Taipei Medical University, Taipei 110, Taiwan

**Keywords:** chronic kidney disease, cohort studies, diabetes mellitus, influenza vaccines

## Abstract

Patients with type 2 diabetes mellitus (T2DM) have a higher risk of chronic kidney disease (CKD) due to vascular complications and chronic inflammation. T2DM contributes to a higher risk of mortality and morbidity related to influenza. In Taiwan, influenza vaccination is recommended for patients with T2DM. A previous meta-analysis reported the efficacy of influenza vaccination in reducing hospitalization and mortality in patients with diabetes; however, the renal protective effect of the vaccine remains unclear. This study evaluated whether influenza vaccination could reduce the incidence of CKD and dialysis in patients with T2DM. The study cohort included all patients aged ≥55 years who were diagnosed as having T2DM between 1 January 2000 and 31 December 2012, by using data from Taiwan’s National Health Insurance Research Database. Each patient was followed up with to assess factors associated with CKD. A time-dependent Cox proportional hazard regression model after adjustment for potential confounders was used to calculate the hazard ratio (HR) of CKD in the vaccinated and unvaccinated patients. The study population comprised 48,017 eligible patients with DM; 23,839 (49.7%) received influenza vaccination and the remaining 24,178 (50.3%) did not. The adjusted HRs (aHRs) for CKD/dialysis decreased in the vaccinated patients compared with the unvaccinated patients (influenza season, noninfluenza season, and all seasons: aHRs: 0.47/0.47, 0.48/0.49, and 0.48/0.48, respectively, all *p* < 0.0001). We observed similar protective effects against CKD during the influenza and noninfluenza seasons. Regardless of comorbidities or drug use, influenza vaccination was an independent protective factor. Furthermore, aHRs for CKD/dialysis were 0.71 (0.65–0.77)/0.77 (0.68–0.87), 0.57 (0.52–0.61)/0.69 (0.56–0.70), and 0.30 (0.28–0.33)/0.28 (0.24–0.31) in the patients who received 1, 2–3, and ≥4 vaccinations during the follow-up period, respectively. This population-based cohort study demonstrated that influenza vaccination exerts a dose-dependent and synergistic protective effect against CKD in the patients with T2DM with associated risk factors.

## 1. Introduction

Type 2 diabetes mellitus (T2DM) is a leading cause of chronic kidney disease (CKD) and end-stage renal disease (ESRD) [1,2]. The prevalence of T2DM in adults is 12.3% and 10.1% in the United States and Taiwan, respectively [3,4]. Patients with T2DM have a high risk of microvascular complications, such as nephropathy, neuropathy, and retinopathy [5]. CKD, as a manifestation of nephropathy, develops in approximately 40% of patients with T2DM and contributes to high mortality and morbidity [6]. The prevalence of not only CKD but also ESRD in Taiwan is the highest worldwide [7,8]. Diabetic kidney disease (DKD) is the leading etiology of CKD development in patients with T2DM [1], with numerous overlapping etiologic pathways, including changes in glomerular hemodynamics, interstitial fibrosis, and tubular atrophy [9]. The natural course of DKD often causes irreversible renal function impairment, resulting in the requirement of dialysis and a poor quality of life [9]. According to the United States Renal Data System, DKD is the single strongest predictor of mortality in patients with T2DM [10,11]. Thus, controlling the development of CKD and ESRD in patients with T2DM is essential.

T2DM increases the risk of severe influenza infection that can lead to higher mortality and morbidity [12]. Severe influenza infection can cause renal complications such as acute kidney injury (AKI) and progression to CKD [13,14]. A prospective cohort study demonstrated that the risk of AKI was higher in patients with T2DM and pandemic influenza A (H1N1) infection, with the prevalence of AKI being 16.2% and 9.6% in the AKI and non-AKI groups, respectively [15]. Moreover, the presence of AKI, which is secondary to rhabdomyolysis, was reported in an 84-year-old man with T2DM and influenza A (H3N2) infection [16]. These findings demonstrate the link between influenza infection and renal impairment in the T2DM population. Considering current influenza treatments, seasonal influenza vaccination is beneficial in minimizing the risks of death and hospitalization from influenza complications in patients with T2DM [17]. A meta-analysis reported the efficacy of the vaccines in reducing hospitalization and mortality in the diabetic population [12]. However, whether influenza vaccines can reduce the incidence of CKD and ESRD in patients with T2DM remains unclear.

The influenza vaccination policy in Taiwan has gradually focused on high-risk populations and key spreaders to effectively reduce the number of influenza cases and deaths. Patients with T2DM as a high-risk population for influenza are considered a priority group for vaccination every year in accordance with the recommendations of the Advisory Committee on Immunization Practices [18]. This study investigated whether the protective benefit of influenza vaccination can reduce the incidence of CKD and ESRD in patients with T2DM. We used data from the National Health Insurance (NHI) Research Database (NHIRD) to determine the association among influenza vaccination, CKD incidence, and dialysis rate in patients with T2DM to evaluate whether influenza vaccination provides renal protection in patients with T2DM.

## 2. Materials and Methods

The NHI program, which was launched in 1995, currently provides comprehensive health insurance coverage to 98% of >23 million people. In this study, we used data from the NHIRD. No significant differences in age, sex, or health-care costs were observed between the study sample and all enrollees [19]. Data in the NHIRD that might be used to identify patients or care providers, including medical institutions and physicians, are encrypted before being sent to the National Health Research Institutes for database construction and are further encrypted before being released to researchers. In other words, querying data alone to identify individuals at any level by using this database is not possible. All researchers using the NHIRD and its subsets are required to sign a written agreement declaring that they have no intention of attempting to obtain information that can violate the privacy of patients or care providers [20]. This study was approved by the Joint Institutional Review Board of Taipei Medical University (approval no. TMUJIRB N201804043).

This study screened all patients who were diagnosed as having diabetes mellitus based on the *International Classification of Diseases, Ninth Revision, Clinical Modification* (*ICD-9-CM*) code 250.X and who visited health-care facilities in Taiwan over 13 years (*n* = 151,605) from 1 January 2000 to 31 December 2012. In the first part, we excluded 98,785 patients due to the following reasons: patients had less than 3 inpatient or outpatient visits related to the diagnosis of diabetes mellitus within 2 years (*n* = 51,124), patients were aged <55 years (*n* = 43,742), and patients were diagnosed as having type 1 DM (*n* = 3919). In the second part, we excluded 4803 patients due to the following reasons: patients had any inpatient or outpatient diagnosis related to CKD before the date of cohort entry (*n* = 2312), patients had any inpatient or outpatient diagnosis related to dialysis before the date of cohort entry (*n* = 36), patients had any inpatient or outpatient diagnosis related to renal transplantation before the date of cohort entry (*n* = 4), and patients had already received vaccination within 6 months before the date of cohort entry (*n* = 2451; Figure 1). In addition, a 1-year washout period (2000) was included to ensure that all patients in this cohort had no CKD or dialysis before enrollment.

In Taiwan, influenza vaccination is provided free of charge and has been recommended for high-risk adults aged ≥50 years (i.e., those with type 2 diabetes, chronic liver infection or cirrhosis, cardiovascular disease, or chronic pulmonary diseases) since 1998 [21]. The vaccination status was recognized based on the presence of code V048 or the use of the vaccine (confirmed by drug codes). We determined the presence of the following comorbidities in each patient: hypertension, cerebrovascular diseases, dyslipidemia, heart diseases, hepatitis B virus, hepatitis C virus, cirrhosis, moderate and severe liver disease, and asthma. Moreover, we collected information regarding antidiabetic medications, co-medications, urbanization level, and monthly income. The primary endpoints of our study were the incidence of CKD (*ICD-9-CM* code 585.X) and the requirement of dialysis (NHI procedure codes) in patients with T2DM. All cohorts were followed up until the date of the diagnosis of CKD, dialysis, death, disenrollment from the NHI, or the end of 2012.

## 3. Statistical Analysis

A propensity score (PS) is used to reduce selection bias and estimate the effect of vaccination by accounting for covariates that predict receiving the intervention (vaccine) by using a logistic regression model [22]. Covariates in the main model were adjusted for PSs for age; sex; hypertension; dyslipidemia; cerebrovascular diseases; heart diseases; hepatitis B virus; hepatitis C virus; cirrhosis; moderate and severe liver disease; asthma; the use of insulin and analogs, biguanides, sulfonamides, urea derivatives, alpha-glucosidase inhibitors, thiazolidinediones, dipeptidyl peptidase 4, other blood glucose–lowering drugs, statins, aspirin, angiotensin-converting enzyme inhibitors, and angiotensin receptor blockers; number of antidiabetic medications; urbanization level; and monthly income (Table 1). Categorical variables were compared using the chi-square test to determine the significance of differences between the vaccinated and unvaccinated groups in terms of the relationship among characteristics listed in Table 1. The unvaccinated group served as the reference arm. The hazard ratio (HR) and 95% confidence interval (CI) for the association between influenza vaccination and the risks of CKD and dialysis in patients with T2DM were calculated using Cox proportional hazards regression. To examine the dose effect of influenza vaccination on the incidence of CKD and dialysis, we categorized patients into four groups by vaccination status: unvaccinated and those receiving 1, 2–3, and ≥4 vaccinations, respectively. These data were stratified according to patients’ age, sex, comorbidity, and associated medication use. Sensitivity analysis was performed to evaluate the difference and consistency between the use of influenza vaccination and the risks of CKD and dialysis in patients with T2DM. All statistical analyses were performed using SPSS 22.0 and SAS 9.4 software. A *p*-value of <0.05 indicated statistical significance.

## 4. Results

### 4.1. Comparison of Baseline Characteristics between the Vaccinated and Unvaccinated Groups

Among the 48,017 eligible individuals enrolled in our cohort study, 49.6% (*n* = 23,839) received the influenza vaccination and the remaining 50.3% (*n* = 24,178) did not receive the influenza vaccination. We noted significant differences (*p* < 0.001) in the distributions of age, sex, preexisting medical comorbidities, antidiabetic medications, number of antidiabetic medications, comorbidity-associated medication use, urbanization level, and monthly income between the groups. The vaccinated group included higher proportions of older and female patients and had a higher prevalence of underlying comorbidities, including hypertension, cerebrovascular diseases, coronary artery diseases, and cirrhosis, before PS adjustment. In addition, the vaccinated group tended to receive different types of antidiabetic medication simultaneously. Moreover, they also received comorbidity-associated medication for longer periods (Table 1).

### 4.2. Differences in Risks of CKD and Dialysis between the Vaccinated and Unvaccinated Groups

The incidence rate of CKD was significantly lower in the vaccinated group (adjusted HR: 0.48, 95% CI: 0.45–0.50, *p* < 0.001) than in the unvaccinated group (Table 2). Similar protective effects were observed in both sexes and all age groups, irrespective of influenza seasonality. The adjusted HR for the incidence rate of CKD was 0.45 (95% CI: 0.41–0.50, *p* < 0.001) and 0.44 (95% CI: 0.41–0.47, *p* < 0.001) in the subgroups of the patients aged 55–64 and ≥65 years, respectively. The protective effect of the influenza vaccination was stronger in the women than in the men, with adjusted HRs of 0.45 (95% CI: 0.41–0.49, *p* < 0.001) in the women and 0.50 (95% CI: 0.46–0.54, *p* < 0.001) in the men in the vaccinated group compared with the unvaccinated group.

The dialysis rate was lower in the vaccinated group (adjusted HR: 0.48, 95% CI: 0.44–0.52, *p* < 0.001) than in the unvaccinated group (Table 3). A similar trend was noted in both sexes and all age groups, irrespective of influenza seasonality. The adjusted HRs were 0.44 (95% CI: 0.38–0.50, *p* < 0.001) and 0.47 (95% CI: 0.42–0.53, *p* < 0.001) in the subgroups of the patients aged 55–64 and ≥65 years, respectively. Similar to the previous finding, the protective effect of the influenza vaccination was stronger in the women than in the men. The men had a higher incidence of dialysis than did the women, with an incidence rate of 799.1 (95% CI: 727.9.2–870.3) and 554.0 (95% CI: 506.8–601.2) per 10^5^ person-years for the unvaccinated and vaccinated men, respectively, and 757.6 (95% CI: 693.4–821.8) and 490.6 (95% CI: 451.6–529.7) per 10^5^ person-years for the unvaccinated and vaccinated women, respectively.

### 4.3. Sensitivity Analysis

In the sensitivity analysis, we adjusted for potential confounders that affected the evaluation between influenza vaccination use and the risk of CKD and dialysis in different models. Potential confounders included comorbidities, demographic variables, and socioeconomic status. When stratified according to the total number of vaccinations, Table 4 and Table 5 demonstrated the protective effects of vaccinated patients in the main model and various subgroups.

As shown in Table 4, we observed a significant vaccine protective effect for the reduction of CKD risk in the main model in the patients who received 1, 2–3, and ≥4 vaccinations during the follow-up period, with the adjusted HRs of 0.71 (95% CI: 0.65–0.77, *p* < 0.001), 0.57 (95% CI: 0.52–0.61, *p* < 0.001), and 0.30 (95% CI: 0.28–0.33, *p* < 0.001), respectively. The result exhibited a similar protective effect in the various subgroups (Table 4). In the main model and various subgroups, our study demonstrated a dose-dependent protective effect of CKD risk reduction. Moreover, in the patients who received ≥4 vaccinations, those aged >65 years presented a lower CKD risk than did those aged <65 years, with the adjusted HRs of 0.25 (95% CI: 0.23–0.28, *p* < 0.001) and 0.31 (95% CI 0.27–0.36, *p* < 0.001), respectively.

We found a significant vaccine protective effect for the reduction of dialysis risk in the main model in the patients who received 1, 2–3, and ≥4 vaccinations during the follow-up period, with the adjusted HRs of 0.77 (95% CI: 0.68–0.87, *p* < 0.001), 0.63 (95% CI: 0.56–0.70, *p* < 0.001), and 0.28 (95% CI: 0.24–0.31, *p* < 0.001; Table 5), respectively. Among the various subgroups, the result showed a similar significant vaccine protective effect. In addition, in the main model and various subgroups, we observed a dose-dependent protective effect of dialysis risk reduction (*p* for trend < 0.001). A more powerful trend in the reduction of dialysis risk was observed in the group aged >65, with the adjusted HRs of 0.82 (95% CI: 0.70–0.97, *p* < 0.001) to 0.25 (95% CI: 0.21–0.29; *p* < 0.001) than in the group aged <65 years, with the adjusted HRs of 0.63 (95% CI: 0.52–0.77, *p* < 0.001) to 0.29 (95% CI: 0.23–0.36, *p* < 0.001).

## 5. Discussion

The results of this population-based cohort study revealed that the patients with T2DM who received the influenza vaccination had lower risks of CKD and ESRD. In our study, patients in the vaccinated group were older, had more preexisting medical comorbidities, used comorbidity-associated medications for longer periods, and used more antidiabetic medications than did the unvaccinated group (Table 1). Patients with T2DM as a high-risk population for influenza and mortality are considered a priority group for vaccination every year in Taiwan [18]. The protective effect of vaccination could still be observed in the vaccinated group compared with the unvaccinated group. Moreover, we observed differences in various factors, including both age and sex, between the groups (Table 2 and Table 3). This is the first population-based cohort study to demonstrate that the protective effect of vaccination can reduce the incidence of CKD and ESRD in the patients with T2DM.

Two mechanisms might explain the main finding of decreased CKD and ESRD risks following influenza vaccination in the patients with T2DM. First, acute hyperglycemia episodes mostly occur in patients with severe influenza infection [23]. Infection-induced glycemic deviation can easily enhance acute hyperglycemia in T2DM and thus worsen infection control (known as a vicious circle) [24,25]. Glycemic deviation is hypothesized to play a crucial role in endothelial cell dysfunction and lead to vascular dysfunction [26,27,28,29]. Glycemic deviation stimulates endothelial cells to overproduce cytokines and overexpress adhesion molecules, leading to the uncontrolled extravasation of leukocytes [30,31] and resulting in organ damage, including renal function impairment [32]. An animal model study reported that hyperglycemia and glycemic deviation impaired renal function by causing endothelial cell dysfunction in conduit vessels and affecting renal circulation [33]. However, additional basic studies are warranted to elucidate mechanisms underlying these findings.

Influenza vaccination can prevent CKD progression by preventing influenza-induced renal injury. Various mechanisms, including acute tubular necrosis caused by renal hypoperfusion or rhabdomyolysis and glomerular microthrombosis resulting from disseminated intravascular coagulation might be responsible for the development of influenza A virus-induced renal injury [13]. Watanabe et al. reported that the prevalence of AKI was high in the population with influenza infection; they examined 45 hospitalized children with seasonal influenza A virus infection and observed that 24.4% of them had renal involvement, of whom 11% developed AKI. Influenza-induced rhabdomyolysis has been reported to cause AKI in several case reports [16,34], indicating the temporal relationship between fever spikes and the subsequent increase in the serum creatine kinase level. AKI episodes are associated with a higher risk of advanced CKD in diabetes mellitus, independent of other major risk factors for progression; each episode of AKI doubles the risk [35]. Several observational studies have consistently demonstrated that a substantial proportion of patients with AKI often recovered while progressing to the advanced stages of CKD [36,37]. Mechanisms underlying progression to CKD after AKI are extremely complex [38] and include the effects of systemic and intrarenal hypertension and glomerular hyperfiltration, tubular hypertrophy and atrophy, tubulointerstitial fibrosis, progressive glomerular sclerosis, arteriosclerosis, genetic susceptibility, and disordered humoral responses [39,40]. Among several pathologic processes, endothelial injury, part of tubulointerstitial damage, and vascular dropout may cause tissue hypoxia and ischemia, thus affecting renal cellular function and causing progression to CKD [41]. However, additional studies are warranted to elucidate the precise mechanism underlying these findings.

In our study, we observed a higher incidence rate of dialysis in the men than in the women, regardless of vaccination status. The protective effect of estrogens on women and the damaging effect of testosterone, together with an unhealthy lifestyle, might result in a faster decline in kidney function in men than in women, as indicated in a previous study [42].

Subgroup analyses by age, sex, preexisting medical comorbidities, and antidiabetic medications demonstrated a significant protective effect in the vaccination group. More favorable protection against CKD was observed in all the age groups in the patients who received 2–3 and ≥4 vaccinations than in those who received only one vaccination (Table 4). A similar trend in the reduction of dialysis risk was noted (Table 5). This dose effect observed in our diabetic population can be related to their dysfunction of the immune response because hyperglycemia reduced the effectiveness of influenza vaccination [43]. Moreover, our study population consisted of patients aged >55 years, and only 17%–53% of clinical vaccine efficacy was noted in the older group, possibly due to an impaired serological response and antibody generation [44]. However, we could not determine whether the patients received annual vaccinations for more than 4 years or received ≥4 vaccinations over the observation period. Our findings indicated that the protective effect of vaccination appeared to be related to its cumulative dose effect. The patients who received more antidiabetic medications had a lower dialysis risk when receiving only one vaccination; stricter blood sugar control might enhance the protective effect of the vaccine. However, additional studies are warranted to elucidate these precise mechanisms. Future prospective studies stratifying patients with factors such as comorbidities and HbA1c levels are required to validate the protective effect of vaccination.

This study has some limitations that should be addressed. First, the retrospective nature of this study limits the generalizability of the findings. Prospective randomized controlled trials are required to confirm the present results. Second, the diagnoses of T2DM and CKD as well as vaccination and dialysis status were determined according to *ICD-9-CM* codes, drug codes, and procedure codes in our study. The diagnostic accuracy of the database may be questionable. Third, the NHIRD does not contain factual information on the severity classification of T2DM as indicated by physical activity, HbA1c, alcohol intake, body mass index, substantial proteinuria, and other laboratory data. Moreover, several confounding factors relevant to CKD and ESRD, including substantial proteinuria, body mass index, and other over-the-counter drug use, were not included in our database [45]. Finally, we presumed that all prescribed medications, including antidiabetic medications and co-medications, were consumed by patients to mitigate the effect of noncompliance, which might affect our result.

## 6. Conclusions

This is the first population-based cohort study to investigate the protective effect of influenza vaccination on CKD incidence and progression to ESRD in patients with T2DM. Our study demonstrated that the protective effect of vaccinations reduced the incidence of CKD and dialysis in patients with T2DM aged ≥55 years. Furthermore, this study showed a dose-dependent effect of vaccination. Our study offers support for the vaccination policy of the Taiwanese government. Large prospective clinical trials should be conducted to elucidate underlying mechanisms based on this study.

## Figures and Tables

**Figure 1 jcm-11-04520-f001:**
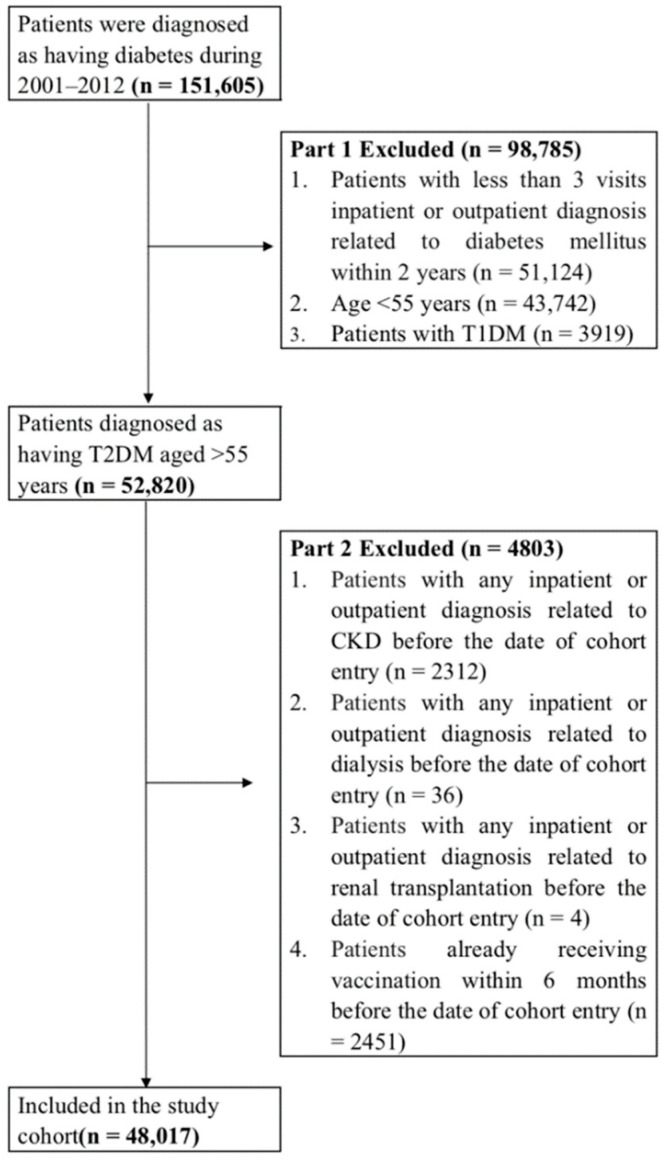
Data Selection Process.

**Table 1 jcm-11-04520-t001:** Characteristics of the Sample Population.

	Whole Cohort (*n* = 48,017)		Unvaccinated (*n* = 24,178)		Vaccinated (*n* = 23,839)		*p*-Value
	*n*	%	*n*	%	*n*	%	
**Age, years (Mean ± SD)**	66.64 (8.12)		64.47 (8.36)		68.85 (7.23)		<0.001
55–64	23,245	48.41	15,259	63.11	7986	33.50	<0.001
65–74	16,669	34.71	5654	23.38	11,015	46.21	
≥75	8103	16.88	3265	13.50	4838	20.29	
**Sex**							
Female	25722	53.57	12,516	51.77	13,206	55.40	<0.001
Male	22,295	46.43	11,662	48.23	10,633	44.60	
**Comorbidities**							
Hypertension	30,218	62.93	14,688	60.75	15,530	65.15	<0.001
Cerebrovascular diseases	8636	17.99	4147	17.15	4489	18.83	<0.001
Dyslipidemia	17,037	35.48	8830	36.52	8207	34.43	<0.001
Heart diseases	18,634	38.81	9172	37.94	9462	39.69	<0.001
Hepatitis B virus	1621	3.38	906	3.75	715	3.00	<0.001
Hepatitis C virus	2618	5.45	1194	4.94	1424	5.97	<0.001
Cirrhosis	2937	6.12	1369	5.66	1568	6.58	<0.001
Moderate and severe liver disease	1118	2.33	564	2.33	554	2.32	0.949
Asthma	6820	14.20	3437	14.22	3383	14.19	0.939
**Antidiabetic medications (ADM)**							
Insulin and analogs	11,918	24.82	4829	19.97	7089	29.74	<0.001
Biguanides	32,915	68.55	15,844	65.53	17,071	71.61	<0.001
Sulfonamides and urea derivatives	30,433	63.38	14,039	58.07	16,394	68.77	<0.001
Other blood glucose-lowering drugs	7963	16.58	3152	13.04	4811	20.18	<0.001
Alpha glucosidase inhibitors	10,416	21.69	4206	17.40	6210	26.05	<0.001
Thiazolidinediones	9359	19.49	3673	15.19	5686	23.85	<0.001
Dipeptidyl peptidase 4	6724	14.00	3039	12.57	3685	15.46	<0.001
**Number of ADM**							
0–1	15,746	32.79	9059	37.47	6687	28.05	<0.001
2–3	17,375	36.19	9295	38.44	8080	33.89	
>3	14,896	31.02	5824	24.09	9072	38.06	
**Co-medications**							
Statin							
<28 days	23,710	49.38	12,663	52.37	11,047	46.34	<0.001
28–365 days	10,143	21.12	5375	22.23	4768	20.00	
>365 days	14,164	29.50	6140	25.39	8024	33.66	
Aspirin							
<28 days	22,807	47.50	13,999	57.90	8808	36.95	<0.001
28–365 days	9406	19.59	4441	18.37	4965	20.83	
>365 days	15,804	32.91	5738	23.73	10,066	42.22	
Angiotensin-converting enzyme inhibitors and Angiotensin receptor blockers							
<28 days	14,916	31.06	9496	39.28	5420	22.74	<0.001
28–365 days	9249	19.26	4994	20.66	4255	17.85	
>365 days	23,852	49.67	9688	40.07	14,164	59.42	
**Level of urbanization**							
Urban	33,968	70.74	18,215	75.34	15,753	66.08	<0.001
Suburban	9453	19.69	4224	17.47	5229	21.93	
Rural	4596	9.57	1739	7.19	2857	11.98	
**Monthly income (NT$)**							
0	5458	11.37	2244	9.28	3214	13.48	<0.001
1–19,200	15,273	31.81	6755	27.94	8518	35.73	
19,200–25,000	13,715	28.56	6082	25.16	7633	32.02	
≥25,001	13,571	28.26	9097	37.63	4474	18.77	

**Table 2 jcm-11-04520-t002:** Risk of CKD in the Unvaccinated and Vaccinated Patients in the Study Cohort.

All Groups (*n* = 48,017)	Unvaccinated (Total Follow-Up of 129,238.4 Person-Years)		Vaccinated (Total Follow-Up of 206,888.8 Person-Years)		Adjusted HR † (95% CI)
	No. of Patients with CKD	Incidence Rate (per 10^5^ Person-Years) (95% CI)	No. of Patients with CKD	Incidence Rate (per 10^5^ Person-Years) (95% CI)	
**Whole cohort**					
Influenza season	1450	1122.0 (1064.2, 1179.7)	1319	637.5 (603.1, 671.9)	0.47 (0.44, 0.51) ***
Noninfluenza season	1308	1012.1 (957.2, 1066.9)	1247	602.7 (569.3, 636.2)	0.48 (0.44, 0.52) ***
All seasons	2758	2134.0 (2054.4, 2213.7)	2566	1240.3 (1192.3, 1288.3)	0.48 (0.45, 0.50) ***
**Age, <65 years** ^a^					
Influenza season	743	846.5 (785.6, 907.4)	350	446.1 (399.3, 492.8)	0.45 (0.39, 0.51) ***
Noninfluenza season	667	759.9 (702.2, 817.6)	322	410.4 (365.6, 455.2)	0.46 (0.40, 0.52) ***
All seasons	1410	1606.4 (1522.6, 1690.3)	672	856.4 (791.7, 921.2)	0.45 (0.41, 0.50) ***
**Age, ≥65 years** ^b^					
Influenza season	707	1705.0 (1579.4, 1830.7)	969	754.5 (707.0, 802.0)	0.44 (0.40, 0.48) ***
Noninfluenza season	641	1545.9 (1426.2, 1665.6)	925	720.3 (673.9, 766.7)	0.45 (0.40, 0.50) ***
All seasons	1348	3250.9 (3077.4, 3424.5)	1894	1474.8 (1408.4, 1541.2)	0.44 (0.41, 0.47) ***
**Female** ^c^					
Influenza season	704	1008.9 (934.3, 1083.4)	636	538.9 (497.0, 580.8)	0.44 (0.40, 0.50) ***
Noninfluenza season	635	910.0 (839.2, 980.8)	633	536.3 (494.6, 578.1)	0.45 (0.40, 0.51) ***
All seasons	1339	1918.9 (1816.1, 2021.6)	1269	1075.2 (1016.1, 1134.4)	0.45 (0.41, 0.49) ***
**Male** ^d^					
Influenza season	746	1254.7 (1164.6, 1344.7)	683	768.6 (710.9, 826.2)	0.50 (0.44, 0.55) ***
Noninfluenza season	673	1131.9 (1046.4, 1217.4)	614	690.9 (636.3, 745.6)	0.51 (0.45, 0.57) ***
All seasons	1419	2386.6 (2262.4, 2510.8)	1297	1459.5 (1380.0, 1538.9)	0.50 (0.46, 0.54) ***

***: *p* < 0.001; ^a^ Total follow-up of 87,773.3 person-years for the unvaccinated group and 78,464.8 person-years for the vaccinated group. ^b^ Total follow-up of 41,465.1 person-years for the unvaccinated group and 128,424.0 person-years for the vaccinated group. ^c^ Total follow-up of 69,781.0 person-years for the unvaccinated group and 118,020.6 person-years for the vaccinated group. ^d^ Total follow-up of 59,457.3 person-years for unvaccinated group and 88,868.2 person-years for the vaccinated group. CI: confidence interval. HR: hazard ratio. † Main model was adjusted for propensity scores for age; sex; hypertension; dyslipidemia; cerebrovascular diseases; heart diseases; hepatitis B virus; hepatitis C virus; cirrhosis; moderate and severe liver disease; asthma; use of insulin and analogs, biguanides, sulfonamides, urea derivatives, alpha glucosidase inhibitors, thiazolidinediones, dipeptidyl peptidase 4, other blood glucose-lowering drugs, antidiabetic medications, statins, aspirin, angiotensin-converting enzyme inhibitors, angiotensin receptor blockers; urbanization level; and monthly income.

**Table 3 jcm-11-04520-t003:** Risk of Dialysis in the Unvaccinated and Vaccinated Patients in the Study Cohort.

All Groups (*n* = 48,017)	Unvaccinated (Total Follow-Up of 131,187.3 Person-Years)		Vaccinated (Total Follow-Up of 219,009.2 Person-Years)		Adjusted HR † (95% CI)
	No. of Patients with Dialysis	Incidence Rate (per 10^5^ Person-Years) (95% CI)	No. of Patients with Dialysis	Incidence Rate (per 10^5^ Person-Years) (95% CI)	
**Whole cohort**					
Influenza season	564	429.9 (394.4, 465.4)	615	280.8 (258.6, 303.0)	0.47 (0.42, 0.53) ***
Noninfluenza season	455	346.8 (315.0, 378.7)	520	237.4 (217.0, 257.8)	0.49 (0.43, 0.56) ***
All seasons	1019	776.8 (729.1, 824.4)	1135	518.2 (488.1, 548.4)	0.48 (0.44, 0.52) ***
**Age, <65 years** ^a^					
Influenza season	327	368.0 (328.1, 407.9)	151	182.9 (153.7, 212.0)	0.38 (0.31, 0.46) ***
Noninfluenza season	239	269.0 (234.9, 303.1)	154	186.5 (157.0, 216.0)	0.52 (0.42, 0.64) ***
All seasons	566	637.0 (584.5, 689.4)	305	369.4 (327.9, 410.8)	0.44 (0.38, 0.50) ***
**Age, ≥65 years** ^b^					
Influenza season	237	559.9 (488.6, 631.2)	464	340.1 (309.1, 371.0)	0.51 (0.43, 0.59) ***
Noninfluenza season	216	510.3 (442.3, 578.4)	366	268.3 (240.8, 295.7)	0.44 (0.37, 0.52) ***
All seasons	453	1070.2 (971.7, 1168.8)	830	608.3 (567.0, 649.7)	0.47 (0.42, 0.53) ***
**Female** ^c^					
Influenza season	297	420.6 (372.7, 468.4)	325	263.1 (234.5, 291.7)	0.44 (0.38, 0.52) ***
Noninfluenza season	238	337.0 (294.2, 379.8)	281	227.5 (200.9, 254.1)	0.46 (0.39, 0.56) ***
All seasons	535	757.6 (693.4, 821.8)	606	490.6 (451.6, 529.7)	0.45 (0.40, 0.51)***
**Male** ^d^					
Influenza season	267	440.8 (388.0, 493.7)	290	303.7 (268.7, 338.6)	0.50 (0.42, 0.60) ***
Noninfluenza season	217	358.3 (310.6, 406.0)	239	250.3 (218.6, 282.0)	0.52 (0.42, 0.63) ***
All seasons	484	799.1 (727.9, 870.3)	529	554.0 (506.8, 601.2)	0.51 (0.45, 0.58) ***

***: *p* < 0.001; ^a^ Total follow-up of 88,860.4 person-years for the unvaccinated group and 82,572.5 person-years for the vaccinated group. ^b^ Total follow-up of 42,326.9 person-years for the unvaccinated group and 136,436.7 person-years for the vaccinated group. ^c^ Total follow-up of 70,621.2 person-years for the unvaccinated group and 123,518.2 person-years for the vaccinated group. ^d^ Total follow-up of 60,566.1 person-years for the unvaccinated group and 95,491.1 person-years for the vaccinated group. CI: confidence interval. HR: hazard ratio. † Main model was adjusted for propensity scores for age; sex; hypertension; dyslipidemia; cerebrovascular diseases; heart diseases; hepatitis B virus; hepatitis C virus; cirrhosis; moderate and severe liver disease; asthma; use of insulin and analogs, biguanides, sulfonamides, urea derivatives, alpha glucosidase inhibitors, thiazolidinediones, dipeptidyl peptidase 4, other blood glucose-lowering drugs, antidiabetic medications, statins, aspirin, angiotensin-converting enzyme inhibitors, angiotensin receptor blockers; urbanization level; and monthly income.

**Table 4 jcm-11-04520-t004:** Sensitivity Analysis of the Adjusted HRs for Vaccination in the Risk Reduction of CKD in All Seasons.

	Unvaccinated	Vaccinated			*p*-Value for Trend
		1	2–3	≥4	
	Adjusted HR (95% CI)	Adjusted HR (95% CI)	Adjusted HR (95% CI)	Adjusted HR (95% CI)	
**Main model †**	1.00	0.71 (0.65, 0.77) ***	0.57 (0.52, 0.61) ***	0.30(0.28, 0.33) ***	<0.001
**Subgroup effects**					
Age, years					
<65	1.00	0.58 (0.51, 0.67) ***	0.51 (0.45,0.59) ***	0.31 (0.27, 0.36) ***	<0.001
≥65	1.00	0.71 (0.64, 0.78) ***	0.52 (0.47,0.57) ***	0.25 (0.23, 0.28) ***	<0.001
Sex					
Female	1.00	0.72 (0.64, 0.80) ***	0.54 (0.48, 0.60) ***	0.28 (0.25, 0.31) ***	<0.001
Male	1.00	0.70 (0.63, 0.79) ***	0.60 (0.54, 0.67) ***	0.33 (0.29, 0.37) ***	<0.001
Hypertension					
No	1.00	0.64 (0.55, 0.74) ***	0.55 (0.48, 0.63) ***	0.32 (0.28, 0.36) ***	<0.001
Yes	1.00	0.74 (0.67, 0.81) ***	0.57 (0.52, 0.63) ***	0.29 (0.27, 0.32) ***	<0.001
Cerebrovascular diseases					
No	1.00	0.72 (0.66, 0.78) ***	0.58 (0.53, 0.63) ***	0.32 (0.29, 0.35) ***	<0.001
Yes	1.00	0.67 (0.56, 0.80) ***	0.51 (0.43, 0.61) ***	0.23 (0.19, 0.29) ***	<0.001
Dyslipidemia					
No	1.00	0.71 (0.64, 0.78) ***	0.57 (0.52, 0.63) ***	0.31 (0.28, 0.35) ***	<0.001
Yes	1.00	0.71 (0.62, 0.82) ***	0.55 (0.48, 0.63) ***	0.28 (0.24, 0.32) ***	<0.001
Heart diseases					
No	1.00	0.68 (0.61, 0.75) ***	0.57 (0.52, 0.63) ***	0.32 (0.29, 0.35) ***	<0.001
Yes	1.00	0.75 (0.66, 0.85) ***	0.55 (0.48, 0.62) ***	0.28 (0.24, 0.32) ***	<0.001
Asthma					
No	1.00	0.70 (0.64, 0.76) ***	0.57 (0.53, 0.62) ***	0.31 (0.28, 0.34) ***	<0.001
Yes	1.00	0.78 (0.62, 0.97) *	0.51 (0.41, 0.64) ***	0.27 (0.21, 0.34) ***	<0.001
Insulin and analogs					
No (<28 days)	1.00	0.72 (0.65, 0.79) ***	0.56 (0.51, 0.61) ***	0.29 (0.26, 0.32) ***	<0.001
Yes (≥28 days)	1.00	0.71 (0.61, 0.81) ***	0.61 (0.53, 0.69) ***	0.35 (0.30, 0.40) ***	<0.001
Biguanides					
No (<28 days)	1.00	0.61 (0.53, 0.70) ***	0.46 (0.40, 0.53) ***	0.22 (0.19, 0.26) ***	<0.001
Yes (≥28 days)	1.00	0.77 (0.70, 0.85) ***	0.63 (0.57, 0.69) ***	0.35 (0.32, 0.38) ***	<0.001
Sulfonamides, urea derivatives					
No (< 28 days)	1.00	0.61 (0.52, 0.71) ***	0.40 (0.34, 0.46) ***	0.23 (0.19, 0.27) ***	<0.001
Yes (≥28 days)	1.00	0.76 (0.70, 0.84) ***	0.65 (0.60, 0.71) ***	0.34 (0.31, 0.38) ***	<0.001
Alpha glucosidase inhibitors					
No (<28 days)	1.00	0.68 (0.62, 0.74) ***	0.53 (0.48, 0.58) ***	0.27 (0.25, 0.30) ***	<0.001
Yes (≥28 days)	1.00	0.85 (0.72, 0.99) *	0.72 (0.62, 0.83) ***	0.42 (0.36, 0.48) ***	<0.001
Thiazolidinediones					
No (<28 days)	1.00	0.69 (0.63, 0.76) ***	0.53 (0.48, 0.58) ***	0.27 (0.25, 0.30) ***	<0.001
Yes (≥28 days)	1.00	0.79 (0.67, 0.93) **	0.72 (0.62, 0.84) ***	0.42 (0.37, 0.49) ***	<0.001
Dipeptidyl peptidase 4 inhibitor					
No (<28 days)	1.00	0.68 (0.63, 0.74) ***	0.54 (0.50, 0.59) ***	0.28 (0.26, 0.30) ***	<0.001
Yes (≥28 days)	1.00	0.97 (0.70, 1.34)	0.83 (0.62, 1.12)	0.69 (0.54, 0.90)**	0.004
Other blood glucose lowering drugs					
No (<28 days)	1.00	0.71 (0.65, 0.78) ***	0.57 (0.52, 0.62) ***	0.28 (0.26, 0.31) ***	<0.001
Yes (≥28 days)	1.00	0.72 (0.60, 0.85) ***	0.58 (0.50, 0.68) ***	0.38 (0.32, 0.44) ***	<0.001
Number of Antidiabetes medications					
0–1	1.00	0.62 (0.53, 0.72) ***	0.42 (0.36, 0.49) ***	0.23 (0.19, 0.27) ***	<0.001
2–3	1.00	0.74 (0.65, 0.85) ***	0.62 (0.55, 0.71) ***	0.29 (0.25, 0.33) ***	<0.001
>3	1.00	0.79 (0.70, 0.90) ***	0.67 (0.59, 0.75) ***	0.40 (0.35, 0.45) ***	<0.001

*: *p* < 0.05; **: *p* < 0.01; ***: *p* < 0.001; † Main model was adjusted for propensity scores for age; sex; hypertension; dyslipidemia; cerebrovascular diseases; heart diseases; hepatitis B virus; hepatitis C virus; cirrhosis; moderate and severe liver disease; asthma; use of insulin and analogs, biguanides, sulfonamides, urea derivatives, alpha glucosidase inhibitors, thiazolidinediones, dipeptidyl peptidase 4, other blood glucose–lowering drugs, antidiabetic medications, statins, aspirin, angiotensin-converting enzyme inhibitors, angiotensin receptor blockers; urbanization level; and monthly income.

**Table 5 jcm-11-04520-t005:** Sensitivity Analysis of Adjusted HRs for Vaccination in the Risk Reduction of Dialysis in All Seasons.

	Unvaccinated	Vaccinated			*p*-Value for Trend
		1	2–3	≥4	
	Adjusted HR (95% CI)	Adjusted HR (95% CI)	Adjusted HR (95% CI)	Adjusted HR (95% CI)	
**Main model †**	1.00	0.77 (0.68, 0.87) ***	0.63 (0.56, 0.70) ***	0.28 (0.24, 0.31) ***	<0.001
**Subgroup effects**					
Age, years					
<65	1.00	0.63 (0.52, 0.77) ***	0.47 (0.38, 0.58) ***	0.29 (0.23, 0.36) ***	<0.001
≥65	1.00	0.82 (0.70, 0.97) *	0.66 (0.58, 0.77) ***	0.25 (0.21, 0.29) ***	<0.001
Sex					
Female	1.00	0.73 (0.62, 0.87) ***	0.59 (0.51, 0.69) ***	0.28 (0.23, 0.33) ***	<0.001
Male	1.00	0.82 (0.68, 0.98) *	0.68 (0.57, 0.80) ***	0.28 (0.23, 0.33) ***	<0.001
Hypertension					
No	1.00	0.64 (0.51, 0.81) ***	0.63 (0.51, 0.76) ***	0.27 (0.21, 0.34) ***	<0.001
Yes	1.00	0.82 (0.71, 0.95) **	0.62 (0.54, 0.71) ***	0.28 (0.24, 0.32) ***	<0.001
Cerebrovascular diseases					
No	1.00	0.75 (0.66, 0.87) ***	0.65 (0.57, 0.73) ***	0.29 (0.25, 0.33) ***	<0.001
Yes	1.00	0.80 (0.61, 1.05)	0.54 (0.41, 0.70) ***	0.22 (0.16, 0.30) ***	<0.001
Dyslipidemia					
No	1.00	0.72 (0.62, 0.83) ***	0.63 (0.55, 0.72) ***	0.28 (0.24, 0.32) ***	<0.001
Yes	1.00	0.90 (0.72, 1.11)	0.63 (0.51, 0.78) ***	0.28 (0.23, 0.36) ***	<0.001
Heart diseases					
No	1.00	0.74 (0.63, 0.86) ***	0.62 (0.53, 0.71) ***	0.28 (0.24, 0.33) ***	<0.001
Yes	1.00	0.82 (0.67, 1.01)	0.64 (0.53, 0.78) ***	0.27 (0.22, 0.33) ***	<0.001
Asthma					
No	1.00	0.77 (0.67, 0.87) ***	0.63 (0.56, 0.71) ***	0.27 (0.24, 0.31) ***	<0.001
Yes	1.00	0.82 (0.55, 1.23)	0.68 (0.47, 0.98) *	0.32 (0.21, 0.48) ***	<0.001
Insulin and analogs					
No (<28 days)	1.00	0.86 (0.72, 1.03)	0.72 (0.61, 0.85) ***	0.26 (0.22, 0.32) ***	<0.001
Yes (≥28 days)	1.00	0.69 (0.58, 0.82) ***	0.57 (0.48, 0.66) ***	0.32 (0.27, 0.37) ***	<0.001
Biguanides					
No (<28 days)	1.00	0.79 (0.62, 1.01)	0.58 (0.46, 0.74) ***	0.23 (0.17, 0.30) ***	<0.001
Yes (≥28 days)	1.00	0.77 (0.67, 0.89) ***	0.66 (0.57, 0.75) ***	0.30 (0.26, 0.35) ***	<0.001
Sulfonamides, urea derivatives					
No (<28 days)	1.00	0.84 (0.63, 1.11)	0.62 (0.47, 0.82) ***	0.26 (0.19, 0.36) ***	<0.001
Yes (≥28 days)	1.00	0.77 (0.67, 0.88) ***	0.65 (0.57, 0.73) ***	0.29 (0.26, 0.33) ***	<0.001
Alpha glucosidase inhibitors					
No (<28 days)	1.00	0.77 (0.66, 0.89) ***	0.64 (0.56, 0.73) ***	0.28 (0.24, 0.33) ***	<0.001
Yes (≥28 days)	1.00	0.78 (0.63, 0.98) *	0.61 (0.49, 0.75) ***	0.27 (0.22, 0.34) ***	<0.001
Thiazolidinediones					
No (<28 days)	1.00	0.69 (0.63, 0.76) ***	0.53 (0.48, 0.58) ***	0.27 (0.25, 0.30) ***	<0.001
Yes (≥28 days)	1.00	0.79 (0.67, 0.93) **	0.72 (0.62, 0.84) ***	0.42 (0.37, 0.49) ***	<0.001
Dipeptidyl peptidase 4 inhibitor					
No (<28 days)	1.00	0.73 (0.64, 0.82) ***	0.61 (0.54, 0.68) ***	0.26 (0.23, 0.30) ***	<0.001
Yes (≥28 days)	1.00	1.29 (0.77, 2.17)	0.73 (0.43, 1.25)	0.54 (0.33, 0.87) *	0.004
Other blood glucose lowering drugs					
No (<28 days)	1.00	0.76 (0.65, 0.88) ***	0.68 (0.59, 0.77) ***	0.28 (0.24, 0.32) ***	<0.001
Yes (≥28 days)	1.00	0.79 (0.64, 0.98) *	0.54 (0.44, 0.67) ***	0.29 (0.24, 0.36) ***	<0.001
Number of Antidiabetes medications					
0–1	1.00	0.86 (0.64, 1.16)	0.66 (0.50, 0.88) **	0.26 (0.19, 0.36) ***	<0.001
2–3	1.00	0.76 (0.62, 0.94) *	0.69 (0.57, 0.83) ***	0.29 (0.24, 0.36) ***	<0.001
>3	1.00	0.78 (0.65, 0.92) **	0.61 (0.52, 0.72) ***	0.30 (0.25, 0.36) ***	<0.001

*: *p* < 0.05; **: *p* < 0.01; ***: *p* < 0.001; † Main model was adjusted for propensity scores for age; sex; hypertension; dyslipidemia; cerebrovascular diseases; heart diseases; hepatitis B virus; hepatitis C virus; cirrhosis; moderate and severe liver disease; asthma; use of insulin and analogs, biguanides, sulfonamides, urea derivatives, alpha glucosidase inhibitors, thiazolidinediones, dipeptidyl peptidase 4, other blood glucose-lowering drugs, antidiabetic medications, statins, aspirin, angiotensin-converting enzyme inhibitors, angiotensin receptor blockers; urbanization level; and monthly income.

## Data Availability

The data presented in this study are available on request from the corresponding authors.

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
