# Peer review of "Effect of Influenza Vaccination on the Reduction of the Incidence of Chronic Kidney Disease and Dialysis in Patients with Type 2 Diabetes Mellitus"

_jcm, 2022, doi:10.3390/jcm11154520_

Round 1
Reviewer 1 Report
The study by Sung et. al. on protective effect of influenza vaccination against increasing incidence of chronic kidney disease in type 2 diabetic patients provides a comprehensive prospective analysis. The authors have performed a thorough analysis of the data curated and provide a good description of their methodology. Although I like that all the analysis is presented (in tables), the authors could do a better presentation of their data especially in table 4 (main model) as reading data in tabular format can get away from the message. The discussion is well written and the authors do a good job in addressing the limitations of the study.
Author Response
Please see the attachment!
Sincerely,
Dr. Liu, Shih-Hao

Reviewer 2 Report
Li-Chin Sung et al. have carried out a retrospective observational clinical study with which they have been able to conclude that the influenza vaccine, administered to type 2 diabetic patients, is capable of significantly reducing the incidence of CKD development and the need for dialysis. It is a work with a good scientific quality that includes a concise and precise introduction, a good description of the methodology, an extensive, detailed and well-argued discussion section and a very appropriate limitations subsection. However, I have to point out some important aspects that need to be improved (mainly in the Results section) before its final publication:
- Figure 1: I recommend increasing the font size (it is difficult to read).
- Line 142: Has it been taken into account that sometimes it is necessary to carry out the Fisher test and not the chi square test?
- I think it is more correct to write p-value than P value (both in the text and in the tables).
- In Table 1... have the statistical analyzes really been carried out correctly? I am quite surprised that in some parameters that seem so similar in the vaccinated and unvaccinated groups, they provide such low p-values... I recommend reviewing the statistical analyzes of this table.
- Line 169: I think that the title of this section does not represent at all what is going to be described later [in the text the most important thing that appears is the incidence of appearance of CKD in both groups (vaccinated and not vaccinated), not I think the most important are the differences in sex and age].
- Why does "No. of Patients With Cancer" appear in Tables 2 and 3?
Author Response

(The authors gave the same response as above.)
